# Predictors of Response to Hydroxyurea and Switch to Ruxolitinib in HU-Resistant Polycythaemia VERA Patients: A Real-World PV-NET Study

**DOI:** 10.3390/cancers15143706

**Published:** 2023-07-21

**Authors:** Francesca Palandri, Elena Rossi, Giuseppe Auteri, Massimo Breccia, Simona Paglia, Giulia Benevolo, Elena M. Elli, Francesco Cavazzini, Gianni Binotto, Alessia Tieghi, Mario Tiribelli, Florian H. Heidel, Massimiliano Bonifacio, Novella Pugliese, Giovanni Caocci, Monica Crugnola, Francesco Mendicino, Alessandra D’Addio, Simona Tomassetti, Bruno Martino, Nicola Polverelli, Sara Ceglie, Camilla Mazzoni, Rikard Mullai, Alessia Ripamonti, Bruno Garibaldi, Fabrizio Pane, Antonio Cuneo, Mauro Krampera, Gianpietro Semenzato, Roberto M. Lemoli, Nicola Vianelli, Giuseppe A. Palumbo, Alessandro Andriani, Michele Cavo, Roberto Latagliata, Valerio De Stefano

**Affiliations:** 1Istituto di Ematologia “Seràgnoli”, IRCCS Azienda Ospedaliero-Universitaria di Bologna, 40138 Bologna, Italy; giuseppe.auteri2@unibo.it (G.A.); camilla.mazzoni@studio.unibo.it (C.M.); nicola.vianelli@unibo.it (N.V.); michele.cavo@unibo.it (M.C.); 2Section of Hematology, Department of Radiological and Hematological Sciences, Catholic University School of Medicine, 00168 Rome, Italy; elena.rossi@policlinicogemelli.it (E.R.); saraceglie@hotmail.it (S.C.); valerio.destefano@unicatt.it (V.D.S.); 3Institute of Hematology, Fondazione Policlinico Universitario A. Gemelli IRCCS, 00168 Rome, Italy; 4Dipartimento di Medicina Specialistica, Diagnostica e Sperimentale, Università di Bologna, 40126 Bologna, Italy; simona.paglia2@unibo.it; 5Division of Cellular Biotechnologies and Hematology, University Sapienza, 00161 Rome, Italy; breccia@bce.uniroma1.it; 6Division of Hematology, Città della Salute e della Scienza Hospital, 10126 Torino, Italy; gbenevolo@cittadellasalute.to.it; 7Hematology Division, San Gerardo Hospital, ASST Monza, 20900 Monza, Italy; elena.elli@libero.it (E.M.E.); alessia.ripamonti@outlook.it (A.R.); 8Division of Hematology, University of Ferrara, 44121 Ferrara, Italy; cvzfnc@unife.it (F.C.); antonio.cuneo@unife.it (A.C.); 9Unit of Hematology and Clinical Immunology, University of Padova, 35020 Padova, Italy; gianni.binotto@unipd.it (G.B.); g.semenzato@unipd.it (G.S.); 10Department of Hematology, Azienda USL-IRCCS di Reggio Emilia, 42122 Reggio Emilia, Italy; alessia.tieghi@ausl.re.it; 11Division of Hematology and BMT, Azienda Sanitaria Universitaria Integrata di Udine, 33100 Udine, Italy; mario.tiribelli@uniud.it (M.T.); mullai.rikard@gmail.com (R.M.); 12Innere Medicine C, Universitätsmedizin Greifswald, 17475 Greifswald, Germany; florian.heidel@uni-greifswald.de; 13Hematology and Bone Marrow Transplant Unit, Section of Biomedicine of Innovation, Department of Engineering for Innovative Medicine, University of Verona, 37134 Verona, Italy; massimiliano.bonifacio@univr.it (M.B.); mauro.krampera@univr.it (M.K.); 14Department of Clinical Medicine and Surgery, Hematology Section, University of Naples “Federico II”, 80131 Naples, Italy; novypugliese@yahoo.it (N.P.); fabrizio.pane@unina.it (F.P.); 15Hematology Unit, Department of Medical Sciences, University of Cagliari, 09124 Cagliari, Italy; giovanni.caocci@unica.it; 16Division of Hematology, Azienda Ospedaliero-Universitaria di Parma, 43126 Parma, Italy; mcrugnola@ao.pr.it; 17Unit of Hematology, Hospital of Cosenza, 87100 Cosenza, Italy; dott.mendicino.ematologiacs@gmail.com; 18Division of Hematology, Onco-Hematologic Department, AUSL della Romagna, 47923 Ravenna, Italy; alessandra.daddio@auslromagna.it; 19Hematology Unit, Infermi Hospital Rimini, 47923 Rimini, Italy; simona.tomassetti@auslromagna.it; 20Division of Hematology, Azienda Ospedaliera ‘Bianchi Melacrino Morelli’, 89133 Reggio Calabria, Italy; brunmartin@libero.it; 21Unit of Blood Diseases and Stem Cell Transplantation, ASST Spedali Civili di Brescia, 25123 Brescia, Italy; nicola.polverelli@unibs.it; 22Postgraduate School of Hematology, University of Catania, 90121 Catania, Italy; brunga93@gmail.com; 23Clinic of Hematology, Department of Internal Medicine (DiMI), University of Genova, 16126 Genova, Italy; roberto.lemoli@unige.it; 24IRCCS Policlinico San Martino, 16132 Genova, Italy; 25Department of Scienze Mediche, Chirurgiche e Tecnologie Avanzate “G.F. Ingrassia”, University of Catania, 95123 Catania, Italy; palumbo.gam@gmail.com; 26Villa Betania Hospital, 00165 Roma, Italy; alessandro.andriani1@tin.it; 27Hematology Unit, Ospedale Belcolle, 01100 Viterbo, Italy; rob.lati@libero.it

**Keywords:** myeloproliferative neoplasms, polycythemia vera, hydroxyurea, ruxolitinib

## Abstract

**Simple Summary:**

The prognostic relevance of a patient achieving complete response to hydroxyurea, the predictors of response, and patients’ triggers for switching to ruxolitinib are uncertain. A retrospective, real-world analysis was performed on 563 polycythemia vera patients treated with hydroxyurea for ≥12 months during an observational “PV-NET” Italian study. We investigated factors associated with a complete response to hydroxyurea and outcomes of the 397 poor responders to hydroxyurea according to whether they subsequently received ruxolitinib (*n* = 114) or continued hydroxyurea (*n* = 283). The results suggest that many PV patients receive underdosed hydroxyurea, leading to lower response and toxicity rates. In addition, many patients continued hydroxyurea despite a poor clinical or hematological response; however, splenomegaly and other symptoms were the main drivers of an early switch. Better HU management, standardization of the criteria for and timing of responses to HU, and adequate intervention in poor responders should be advised.

**Abstract:**

In polycythemia vera (PV), the prognostic relevance of an ELN-defined complete response (CR) to hydroxyurea (HU), the predictors of response, and patients’ triggers for switching to ruxolitinib are uncertain. In a real-world analysis, we evaluated the predictors of response, their impact on the clinical outcomes of CR to HU, and the correlations between partial or no response (PR/NR) and a patient switching to ruxolitinib. Among 563 PV patients receiving HU for ≥12 months, 166 (29.5%) achieved CR, 264 achieved PR, and 133 achieved NR. In a multivariate analysis, the absence of splenomegaly (*p* = 0.03), pruritus (*p* = 0.002), and a median HU dose of ≥1 g/day (*p* < 0.001) remained associated with CR. Adverse events were more frequent with a median HU dose of ≥1 g/day. Overall, 283 PR/NR patients (71.3%) continued HU, and 114 switched to ruxolitinib. In the 449 patients receiving only HU, rates of thrombosis, hemorrhages, progression, and overall survival were comparable among the CR, PR, and NR groups. Many PV patients received underdosed HU, leading to lower CR and toxicity rates. In addition, many patients continued HU despite a PR/NR; however, splenomegaly and other symptoms were the main drivers of an early switch. Better HU management, standardization of the criteria for and timing of responses to HU, and adequate intervention in poor responders should be advised.

## 1. Introduction

Polycythemia vera (PV) is a Philadelphia-negative chronic myeloproliferative neoplasm (MPN) characterized by the clonal expansion of an erythrocyte mass due to mutations in the *JAK2* gene (V617F and exon 12) that cause hyperactivation of JAK-STAT signaling. It is clinically burdened by thrombotic complications, systemic symptoms, progressive splenomegaly, and a risk of evolution into post-PV myelofibrosis (PPV-MF) and blast phase (BP) [1,2,3,4].

Hydroxyurea (HU) is currently the most used cytoreductive therapy for PV patients at high risk of thrombosis, with most patients achieving adequate control of the disease with acceptable tolerance. However, many patients may only obtain a poor response to HU or develop drug-related toxicities during therapy [5,6,7,8,9]. 

Standardized criteria for clinico-hematological responses to HU and a unified definition of resistance/intolerance to HU in PV have been proposed by the European Leukemia Net (ELN) [10,11]. These indications have become particularly relevant after the approval of ruxolitinib, a *JAK1/JAK2* inhibitor, in cases of intolerance or resistance to HU. In the pivotal RESPONSE and RESPONSE-2 studies, ruxolitinib was superior to the best available therapy at controlling hematocrit and improving splenomegaly and other symptoms after HU failure, achieving long-lasting responses in many cases [12,13,14,15,16]. 

The predictive factors of complete response (CR) to HU and the prognostic relevance of its achievement have yet to be defined. Additionally, the impact of different types of suboptimal response (i.e., inadequate control of hematocrit/leukocytosis/thrombocytosis/ or persistent/progressive splenomegaly/symptoms) on patients switching to ruxolitinib is uncertain. In 563 PV patients treated with HU for at least 12 months, we aimed to (1) identify clinical/laboratory characteristics associated with the achievement of CR to HU, (2) investigate whether the type of poor response to HU may influence a patient’s decision to switch to ruxolitinib, and (3) evaluate whether achieving CR to HU may improve outcome parameters, including thromboses, hemorrhages, progression to PPV-MF/BP, and survival.

## 2. Materials and Methods

### 2.1. Patients and Study Design

This observational, retrospective cohort study (PV-NET) was promoted by the IRCCS Azienda Ospedaliero-Universitaria S. Orsola-Malpighi, Bologna, Italy. The study involves 934 PV patients diagnosed between January 1985 and December 2020 in 22 academic hematology centers (Appendix B). After approval from each IRB, the centers collectively submitted diagnostic and follow-up information. The totality of medical files from each center was reported via data input into an electronic database that was developed to label all study data with an alphanumeric code after the de-identification of patients to protect personal privacy. 

The data collected included patient demographics, medications, clinical/laboratory tests at diagnosis and during follow-up, type of PV therapy, death, and causes of death. All information about concomitant diseases, body mass index (BMI), Charlson Comorbidity index (CCI), cardiovascular risk factors (CVRF), thrombosis history, and drug usage was recorded in each case history and, thereafter, used for retrospective evaluation. Any treatment decision, including the use of phlebotomies and antiplatelet drugs, was at the physician’s discretion and independent from participation in this study.

After the first data entry, follow-up information was validated with revision of the clinical data, and specific queries were addressed to the participating centers in cases of inconsistent data. All patients were followed until death or the data cut-off date (October 2021).

### 2.2. Definitions

PV was diagnosed according to the WHO 2016 classification [17]. In patients diagnosed before 2016, marrow biopsies were internally reviewed to adhere the data to current criteria [17]. In patients without a bone marrow histology, PV diagnosis was based on the presence of elevated hemoglobin levels (i.e., >18.5 and 16.5 g/dL in males and females, respectively), the *JAK2* mutation, and low serum erythropoietin. Conventional criteria were used for the diagnosis of blast phase (BP) [18] or PPV-MF [19].

We operatively adopted the European LeukemiaNet 2009 criteria [20] without considering the updated criterion of histologic remission [11], since such a work-up was intended for clinical trials exploring novel drugs and not for a real-life clinical setting. CR to HU (up to 2 g/die) was defined as a hematocrit of <45% without phlebotomy, a platelet count of ≤400 × 10^9^/L, a leukocyte count of ≤10 × 10^9^/L, a normal spleen size, and no disease-related symptoms. A hematocrit of <45% without phlebotomy or response in ≥3 of the other criteria was defined as partial response (PR). No response (NR) was any response that did not satisfy partial response [20]. Patients who never achieved CR to HU at any clinical hematological evaluations after ≥12 months of therapy were defined as poor responders. Patients who switched to ruxolitinib were defined as HU-RUX and patients who continued HU despite a poor response were defined as HU-POOR. 

CVRF included smoking, diabetes, hypertension, dyslipidemia, and being overweight [21]. Overweight was defined as BMI ≥ 25. CCI was evaluated according to the Charlson’s score [22]. Thromboses were defined according to the International Classification of Diseases (9th revision) and graded according to the Common Terminology Criteria for Adverse Events (CTCAE) v4.0.

PV-related symptoms and pruritus at diagnosis and over time were deduced from medical records. The tolerability of HU was evaluated and graded according to CTCAE v 4.0 by the treating hematologist through routine clinical examinations and tests.

### 2.3. Ethical Aspects

This PV-NET study was performed in accordance with the guidelines of the IRBs of the participating centers and the standards of the Helsinki Declaration. For patients currently under follow-up with the experimental center, informed consent was obtained as part of one of the visits in their normal care pathway. For deceased patients, Italian regulations authorized the processing of personal data carried out for scientific research purposes (Gazzetta Ufficiale no. 72 dated 26 March 2012). Therefore, the processing of personal data is considered authorized upon approval of the study by the Ethics Committee. The promoter of this study was the IRCCS Azienda Ospedaliero-Universitaria S. Orsola-Malpighi, Bologna, which obtained approval from the Area Vasta Emilia Centro (AVEC) Ethics Committee (approval file number: 438/2018/Oss/AOUBo). This study was also approved by the local ethics committees of participating centers (protocol code: PV-ARC) and had no commercial support.

### 2.4. Statistical Analyses

Statistical analysis was carried out at the biostatistics laboratory of the MPN Unit, IRCCS Azienda Ospedaliero-Universitaria S. Orsola-Malpighi, Bologna.

Comparisons of quantitative variables between groups of patients were carried out using the Wilcoxon–Mann–Whitney rank-sum test, and associations between categorical variables were tested using the χ^2^ test. In patients with a stable poor response, the index date (ID) was set to 12 months from HU start. The 12-month cut-off was chosen as it represents an adequate time to optimize the dosage and evaluate the efficacy of HU therapy. In HU-CR patients, the index date was set to the time of achievement of CR.

The determination of the most appropriate cut-off value, identified using the Youden index, for the median HU dose in relation to the achievement of CR was based on an ROC (receiver operating characteristic) analysis, with AUC (area under curve) values > 0.70. Multivariable analysis of baseline characteristics associated to CR was carried out using a logistic regression model. Odds Ratios (OR) and 95% Confidence Intervals (95% CI) for the variables associated to CR were specified.

HU-related adverse events were reported as incidence rates (IR). Incidence rates were compared with the exact mid-P estimation method. Events reported for HU-CR (n. 166) and HU-POOR patients (n. 283) occurred during HU therapy, and time was considered from ID to HU discontinuation or last contact. 

The switch to ruxolitinib was considered a time-to-event variable, calculated from the patient’s index date to starting ruxolitinib. Ten poor responders died before ruxolitinib was available and were excluded from the following analyses. The predictors for early switch (within 12 months from the index date to starting ruxolitinib) were identified using the Cox proportional-hazards regression model with an adjustment for left-truncation (delayed entry), while predictors of late switch (>12 months from index date) were identified with the Fine and Gray model, treating early switch as a competing event and adjusting for delayed entry. The proportional hazards assumption held and the Cox regression hazard ratio (HR) estimated from the left-truncated data was an unbiased estimate of the true HR. The proportional hazards assumption was assessed with log-log plots. A variable selection for multivariable analyses was carried out using augmented backward elimination.

The overall survival (OS) was calculated from the index date to the last contact. Univariate comparisons were carried out with log-rank tests. Survivor functions of CR, PR, and NR patients were plotted after the Cox proportional hazards multivariable regression model, adjusting for age at index date (associated with OS in the univariate analysis when comparing CR, PR, and NR patients).

Event-free (events including major thrombosis, hemorrhage, transformation into PPV-MF/BP, and death) survival (EFS) was measured from the index date to the event. Progression-free survival (PFS) was determined from the index date to the date of progression to PPV-MF or BP or censored at the date of death or last follow-up. Both EFS and PMF were estimated using Kaplan–Meier (KM) methods and compared between CR, PR, and NR groups using the log-rank test.

For all tested hypotheses, two-tailed *p*-values < 0.05 were considered significant.

### 2.5. Data Sharing Statement

The datasets used and analyzed during the current study are available from the corresponding author on reasonable request.

## 3. Results

### 3.1. Study Cohort

Among the 934 PV patients included in the PV-NET database, 52, 15, 41 and 212 patients were excluded because their PV diagnosis did not meet the 2016 WHO criteria, because of excess missing data, because their HU therapy was prior to the discovery of the *JAK2^V617F^* mutation (2005), and because they had no or less than 12 months of exposure to HU, respectively. Additionally, 51 patients who never achieved CR to HU were excluded because they received no therapy or only busulfan/interferons after HU discontinuation. Overall, 563 patients received HU for at least 12 months and were included in this analysis (Figure 1).

### 3.2. Efficacy and Safety of Hydroxyurea

HU was used in 98.1% of the patients as a front-line therapy, mainly (87.6%) due to high-risk criteria (age > 60 or previous thrombosis). Median HU exposure was 4.6 years (range 1–14.8), with a total observation time of 3200 patient years. In 506 evaluable patients, median HU dose was 0.5 g/d (range, 0.2–2) and was ≥2 g/d in 2.9% of patients. Only 160 patients (31.6%) received median HU doses of ≥1 g/d.

After the start of HU, 166 (29.5%) patients achieved CR (HU-CR), while 397 (70.5%) always had a poor response, including 264 (46.9%) partial responses (PR) and 133 (23.6%) patients with no response (NR).

Table 1 shows the main patient characteristics of the study cohort according to their response to HU. In the multivariable analysis, an absence of pruritus (OR [95% CI]: 3.23 [1.55–6.75], *p* = 0.002), an absence of palpable splenomegaly (OR [95% CI]: 2.31 [1.10–4.85], *p* = 0.03), and a median HU dose of ≥1 g/d (OR [95% CI]: 4.69 [2.59–8.49], *p* < 0.001) were confirmed to have a significant association with CR. No significant difference in median HU doses was observed between PR and NR patients (*p* = 0.08).

In the 160 patients who received the median HU dose of ≥1 g/d, a *JAK2*^V617F^ variant allele frequency (VAF) of <50% (OR [95% CI]: 2.34 [1.02–5.58], *p* = 0.05) and the absence of palpable spleen (OR [95% CI]: 2.69 [1.02–7.10], *p* = 0.04) were confirmed to have an association with CR. 

At least one HU-related adverse event of a grade ≥2 occurred in 128/563 (22.7%) patients, with an overall incidence rate of 5.8 per 100 patient-years. An increased incidence of adverse events overall (*p* = 0.002), anemia (*p* = 0.03), and skin ulcers (*p* = 0.02) was associated with the median HU dose of ≥1 g/d (Table 2).

### 3.3. Treatment Strategy in Patients with Stable Poor Response

After a median time from the index date of 4.0 years (range 0.1–12.5), 114 (28.7%) poor responders switched to ruxolitinib (HU-RUX). In 50 HU-RUX patients (43.9%), HU was also discontinued due to toxicity. The median ruxolitinib exposure was 1.5 years (range 0.1–6.8), with a total observation of 188.5 patient-years. Conversely, 283 (71.3%) patients continued HU (HU-POOR).

Compared to HU-POOR patients, HU-RUX patients were younger (*p* < 0.001) and more frequently presented with palpable spleen (*p* = 0.004) and pruritus (*p* < 0.001) (Appendix A). 

At 5 years from the index date, the probability of a switch to ruxolitinib was significantly higher in patients with both hematological and spleen/symptom criteria (*p* < 0.001) (Figure 2a). Analogously, patients with NR had a higher probability of RUX switch compared to PR patients (*p* < 0.001) (Figure 2b).

Overall, 24 (21%) out of 114 HU-RUX patients switched to ruxolitinib within 12 months (early switch), while 90 (79%) switched after >12 months from index date (late switch). In the multivariable analysis, splenomegaly (*p* = 0.04) was specifically associated with an early switch (Figure 3a). The need for phlebotomies (*p* = 0.01), the median HU dose of ≥1 g/d (*p* < 0.001), and pruritus (*p* < 0.001) were specifically associated with a late switch (Figure 3b). However, as expected, patients with an early RUX switch more frequently had an index date after 2017, which corresponds to the date of RUX availability in Italy (*p* < 0.001).

### 3.4. Outcome according to Response to Hydroxyurea

Frequency and incidence rates (IR) of events during HU therapy are detailed in Table 3.

Overall, 51 thromboses in 43 patients, 25 hemorrhages in 25 patients, and 43 infections in 35 patients occurred during HU, with an overall incidence rate of 2.33, 1.07, and 2.38 per 100 patient years, respectively. Fifty-two patients had a second primary malignancy during HU with an overall incidence rate of 2.49 per 100 patient years. Finally, 10 patients progressed to BP (IR 0.41 × 100 patient-years), 14 developed a PPV-MF (IR 0.83 × 100 patient-years), and 35 died (IR 1.46 × 100 patient-years). The causes of death and incidence rates for each are reported in Appendix A.

In a time-dependent, multivariable Cox proportional hazards model, achieving an ELN response was not associated with a reduced risk of developing a thrombosis (HR, 1.07; 95% CI, 0.49–2.36, *p* = 0.86). Of note, in this model, a prior thrombosis was associated with the development of a subsequent thrombosis (HR, 2.67; 95% CI, 1.23–5.80, *p* = 0.01). Obtaining an ELN response with HU therapy was not associated with a decreased risk of disease progression into MF/BP (HR,1.06; 95% CI, 0.43–2.60, *p* = 0.90) or a reduced risk of death (HR, 0.94; 95%CI, 0.47–1.87, *p* = 0.86). The only risk factor associated with an increased risk of death was an age ≥65 years (HR, 3.70; 95% CI, 1.29–10.63, *p* = 0.02).

In the 449 patients receiving only HU, OS was not significantly influenced by response (*p* = 0.64) (Figure 4a). Analogously, PFS and EFS were comparable across the three cohorts (respectively, *p* = 0.94 and *p* = 0.95) (Figure 4b,c).

## 4. Discussion

This real-world study investigated the characteristics associated with the achievement of an ELN-defined CR to HU, the prognostic impact of meeting these criteria, and the real-world triggers for poor responders to HU switching to RUX.

First, a baseline absence of splenomegaly and other symptoms as well as a *JAK2*^V617F^ VAF of <50% were the main disease-related features associated with CR, and they may identify PV patients who are more likely to benefit from HU. As in other real-world experiences, a generalized use of low-dose HU was observed, possibly resulting in low rates of CR (29.5%) that were, however, superior to those reported by the Spanish registry (21%) [7,23]. Whether this is a clinical practice to be improved or to be integrated into future definitions remains to be clarified. Indeed, while CR was more frequent in patients receiving the median HU of ≥1 g/day, such a dose was associated with significantly higher toxicity rates, namely anemia and skin ulcers. The optimization of HU doses and a better clinical evaluation and formal assessment of HU-related toxicity may result in a significant improvement in CR rates and are warranted.

Second, many patients continued HU therapy despite a poor response, and ruxolitinib was mostly reserved for truly refractory cases. This was possibly related to several factors: ruxolitinib was not readily available for patients with an older diagnosis, the age of patients with PV is quite high, and comorbidities, logistical problems, and initial unfamiliarity with ruxolitinib on the part of treating hematologists may have further contributed to this therapeutic inertia. Notably, an ongoing need for phlebotomies, which may be suggestive of suboptimal hematocrit control, was generally tolerated during the first two years after evidence of a poor response to HU. This points out that, in real-life practice, the importance of targeting a hematocrit below 45% is still underestimated despite its significant correlation with improved survival [24]. This study was not designed to assess any outcome benefit of switching to ruxolitinib. However, the Majic-PV prospective trial has recently highlighted that, in HU intolerant/resistant patients, event-free survival without major hemorrhage, thrombosis, transformation, or death was superior for both ruxolitinib patients and patients who attained CR within 1 year [25]. Notably, compared to the Majic-PV trial, our study also included a complete resolution of PV-related symptoms among the criteria for CR; additionally, we evaluated patients who attained CR one year from the start of HU and maintained such a response at all evaluable timepoints. These differences in response definitions and timings may at least partially explain the different results on the impact of CR on outcome. Additionally, a slight survival advantage was observed in the prospective Response-2 trial in RUX-treated patients compared to the control arm [16]. 

Finally, HU-CR patients had comparable outcomes, including the probability of thromboses, disease progression, and survival, compared with those of partial and non-responders during HU therapy. Particularly, the lack of an association between CR and a reduced thrombosis rate is noteworthy and is likely due to the interaction of multiple factors, difficult to capture in retrospective analysis, in the genesis of thrombotic events. Accordingly, neither hematocrit nor leukocyte and platelet counts were significantly associated with the risk of thrombosis in a recent retrospective analysis [26].

Additionally, previous observations have suggested that ELN criteria are not informative when evaluating new therapies with the primary goal of reducing mortality [6,7,27]. However, it is important to highlight that persistently elevated leukocyte trajectories, rather than single leukocyte evaluations, were recently found to be associated with an increased risk of disease evolution, suggesting that continuous hematological monitoring may provide significant prognostic insights [26].

Due its retrospective nature, this study has some inherent limitations. Indeed, patient selection, uncontrolled drug prescription, inadequate recognition of poor response to HU, and scarce assessment of drug compliance cannot be entirely ruled out. However, the substantial number of included patients, the cooperation of hematology centers dedicated to MPNs, the long follow-up with a median HU exposure of 4.6 years, and the accurate revision of each case history with no cases lost to follow-up may partially compensate for these intrinsic shortcomings.

## 5. Conclusions

Our results emphasize the importance of optimizing HU dosing to achieve CR but also recognize the association between higher HU doses and drug-related toxicity. This finding, together with the lack of an association between CR and reduced thrombotic risk, suggests that the risks and benefits of pursuing CR through higher HU doses should be balanced in an individualized manner.

Additionally, we observed that HU was continued in a large portion of patients with stable suboptimal response. The experiences observed in the Majic-PV study and the response study call attention to the need to recognize poor responders and to improve the overall therapeutic strategy in PV treatment.

The recent ELN 2021 recommendations provide indications for switching from HU to another cytoreduction, extending the criteria of clinical response to HU to include, besides the cell counts and the spleen size, a more detailed evaluation of the constitutional symptoms and the phlebotomies needed [5]. More important, the dose of HU considered for response was downgraded to 1.5 g/day [5]. Considering the scarce effect of the ELN 2009 criteria on clinical outcomse, the introduction of the ELN 2021 recommendations is warranted.

## Figures and Tables

**Figure 1 cancers-15-03706-f001:**
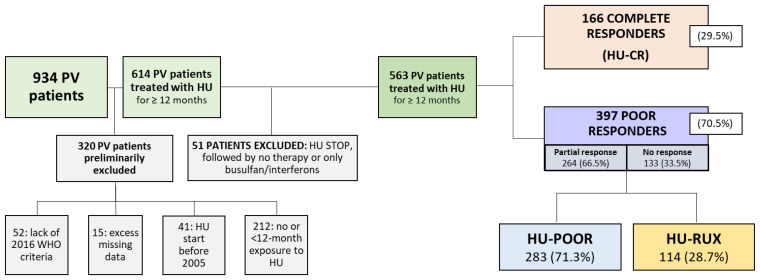
Study flowchart. Number of individuals at each stage of the study.

**Figure 2 cancers-15-03706-f002:**
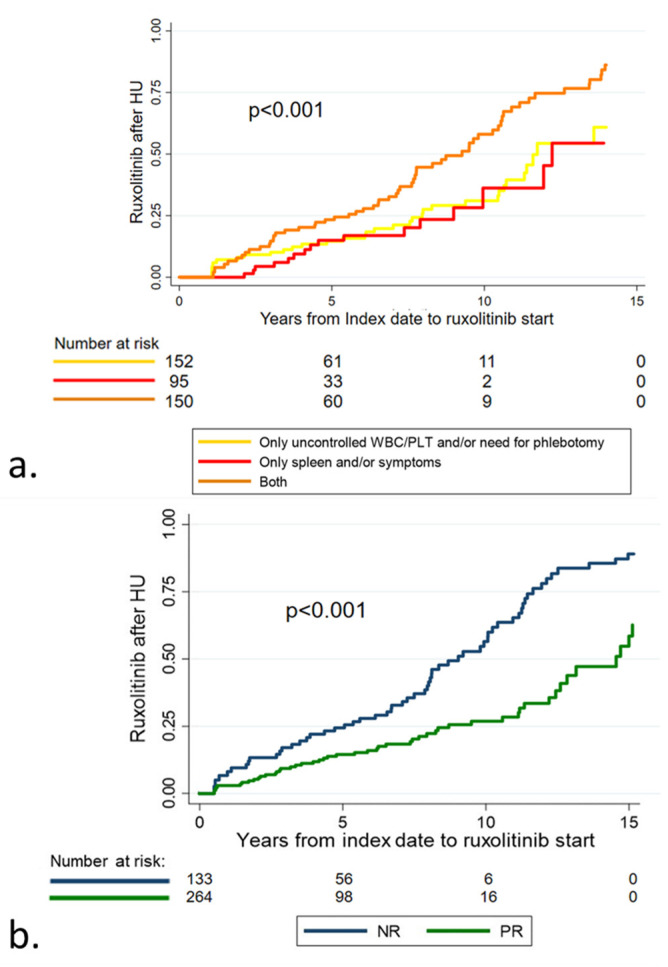
Probability of switch to ruxolitinib according to type of poor response (**a**) and to type of response to hydroxyurea (HU) (**b**) NR: no response. PR: partial response. Overall, 213 (37.8%) patients had uncontrolled myeloproliferation (leukocyte count >10 × 10^9^/L and platelet count 400 × 10^9^/L), 190 (33.7%) needed phlebotomies to keep hematocrit at <45%, and 122 (21.8%) and 172 (30.6%) had persistence or occurrence of palpable splenomegaly or PV-related symptoms, respectively. Poor response mainly consisted in the combination of hematological and spleen/symptom criteria (37.8%) or only hematological criteria (38%).

**Figure 3 cancers-15-03706-f003:**
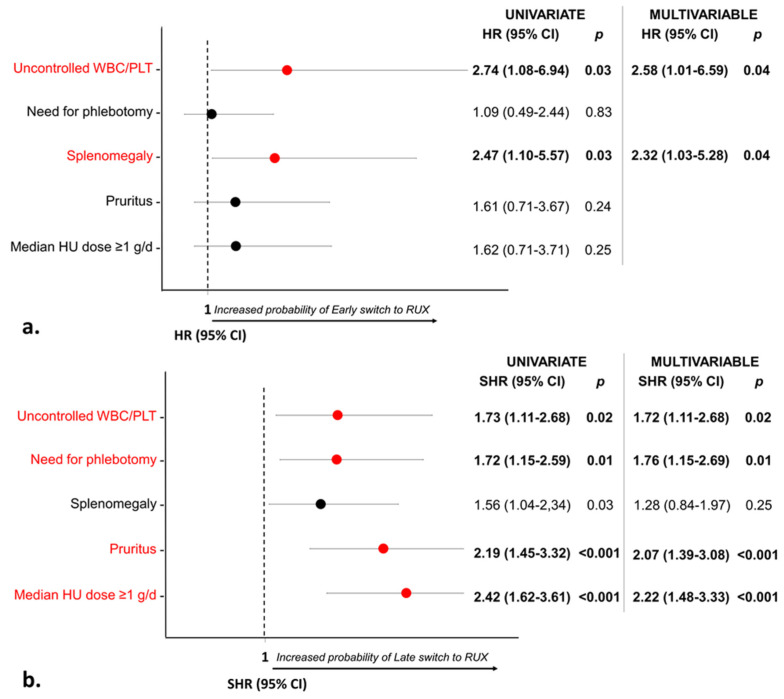
Factors associated with early (**a**) and late (**b**) switch to ruxolitinib. Predictors of early switch to RUX were identified with the Cox proportional hazards regression model, while predictors of late switch to RUX were identified with the Fine and Gray model, treating early switch as a competing event. Both models, from ID to RUX start/last contact, were adjusted for delayed entry. Red dots indicate significant variables in both uni- and multivariate.

**Figure 4 cancers-15-03706-f004:**
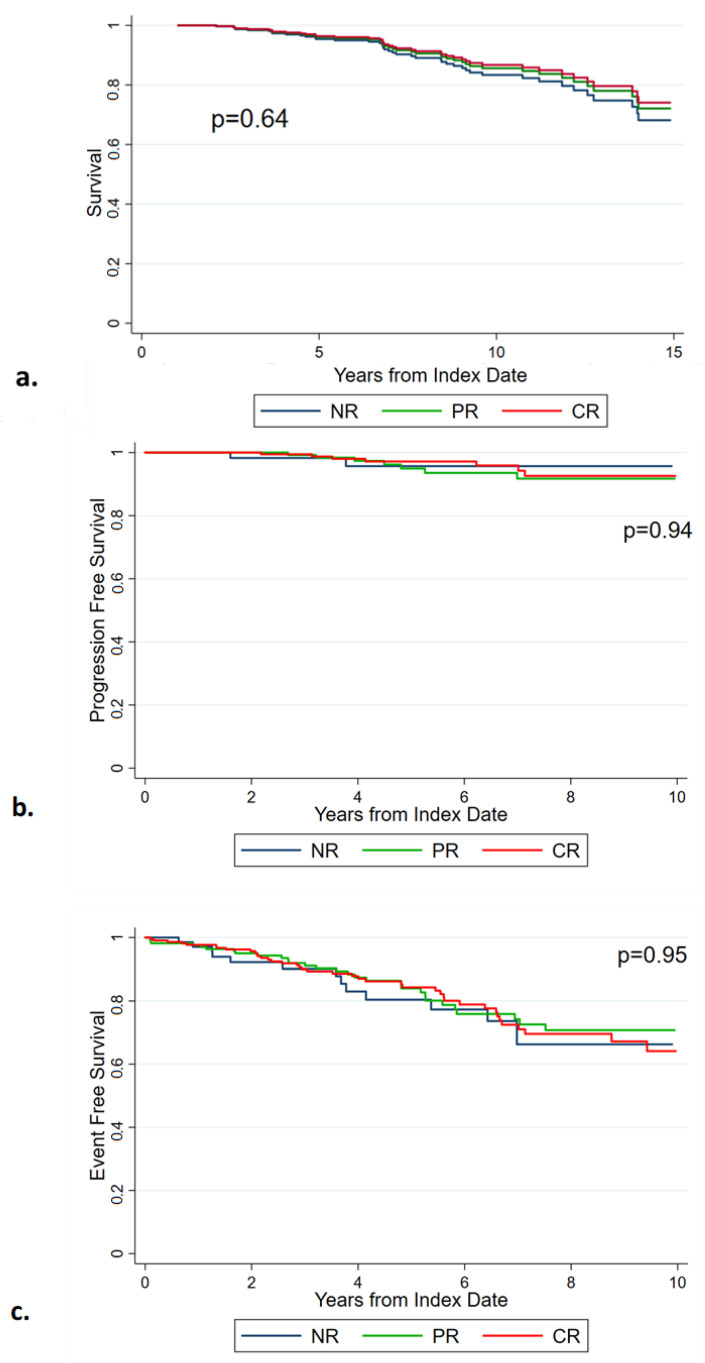
Overall survival (**a**), progression-free survival (**b**), and event-free survival (**c**) of the 449 patients treated with only hydroxyurea, according to response. NR: no response. PR: partial response. CR: complete response.

**Table 1 cancers-15-03706-t001:** Main patient characteristics at diagnosis according to response to hydroxyurea ^1^.

Characteristics	Complete Responders (n. 166)	Poor Responders (n. 397)	*p* Value
Age, median (range), years Age ≥ 65 years	70 (47–87) 116 (69.9%)	65 (21–89) 206 (51.9%)	<0.001 <0.001
Male sex, no. (%)	66 (39.8%)	220 (55.4%)	0.001
*JAK2*^V617F^ VAF ≥ 50%, no. (%) on 365 evaluable	45/114 (39.5%)	135/251 (53.8%)	0.01
Platelet count, median (range), ×10^9^/L	500 (159–1279)	449 (138–1209)	0.004
Leukocytes, median (range), ×10^9^/L	10 (3.3–30.3)	10.1 (1–27.3)	0.70
Hemoglobin, median (range), g/dL			
Male	18.6 (15.8–23.6)	18.7 (12–23.4)	0.59
Female	17.8 (15.3–22)	17.5 (13.2–21.9)	0.09
Hematocrit, median (range), %			
Male	55 (48.9–72.5)	56.3 (38–73)	0.70
Female	54 (47.6–71.7)	54.1 (39–72)	0.99
Palpable spleen, no. (%) of 548 evaluable	26/165 (15.8%)	151/383 (39.4%)	<0.001
Pruritus, no. (%)	28 (17%)	158 (39.9%)	<0.001
BMI ≥ 25, % of 349 evaluable	32/65 (49.2%)	144/284 (50.0%)	0.83
At least one CVRF, no. (%)	129 (77.7%)	312 (78.6%)	0.81
Thromboses pre-/at diagnosis, no. (%)	39 (23.5%)	102 (25.7%)	0.58
Median HU dose, median (range), g/d of 506 evaluable Median HU dose ≥ 1 g/d, no. (%)	0.8 (0.2–2) 59/119 (49.6%)	0.5 (0.2–2) 101/387 (26.17%)	<0.001 <0.001

^1^ HU: hydroxyurea. CVRF: cardiovascular risk factor. Thrombosis history included acute myocardial infarction, transient ischemic attack/stroke, superficial vein thrombosis, major venous thromboembolism, and acute/chronic arterial obstructive disease. Median dose ≥ 1 g/d was determined as the best cut-off value based on ROC analysis (AUC: 0.76).

**Table 2 cancers-15-03706-t002:** Adverse events ^1^ related to hydroxyurea (HU) therapy, according to median HU dose.

Toxicities	HU < 1 g/d (n. 346)		HU ≥ 1 g/d (n. 160)		*p*
	n. (%)	Incidence Rate (per 100 Patient-Years)	n. (%)	Incidence Rate (per 100 Patient-Years)	
**Hematological toxicity**	22 (6.4%)	1.7	26 (16.3%)	4.0	**0.003**
**Anemia**	5 (1.5%)	0.4	9 (5.7%)	1.3	**0.03**
**Thrombocytopenia**	15 (4.3%)	1.2	16 (10%)	2.5	0.09
**Neutropenia**	2 (0.6%)	0.1	1 (0.6%)	0.2	1.0
**Extra-hematological toxicity**	42 (12.1%)	3.1	33 (20.6%)	4.7	0.11
**Skin ulcers**	18 (5.2%)	1.4	20 (12.5%)	2.9	**0.02**
**Oral aftosis**	9 (2.6%)	0.7	4 (2.5%)	0.6	0.81
**Gastro-intestinal disturbances**	4 (1.1%)	0.3	3 (1.9%)	0.4	0.65
**Fever**	2 (0.6%)	0.1	0	0	0.43
**Myalgia**	2 (0.6%)	0.1	0	0	0.43
**Zoster reactivations**	1 (0.3%)	0.1	1 (0.6%)	0.2	0.69
**Non-melanoma skin cancer**	6 (1.7%)	0.4	5 (3.1%)	0.6	0.53
**Overall toxicity**	64 (18.5%)	4.8	59 (36.9%)	8.7	**0.002**

^1^ Only grade ≥2 adverse events have been reported.

**Table 3 cancers-15-03706-t003:** Frequency and incidence rates of events ^1^.

		HU-CR (n.166)		HU-POOR (n.283)		HU-CR versus HU-POOR
**Exposure (patient-years)**		*717.4*		*1182.4*		
**Events**	Total n.	n.	Incidence rate (per 100 patient-years)	n.	Incidence rate (per 100 patient-years)	*p*
**Thromboses** **Arterial** **Venous**	51 24 27	16 9 7	2.36 1.33 1.03	25 11 14	2.23 0.98 1.25	0.47
**Haemorrhages**	25	13	1.45	12	0.84	0.17
**Infections** **Lung** **Mucocutaneous** **Urinary tract** **Herpes zoster** **Herpes simplex** **Gastrointestinal** **Sepsis** **Other**	43 14 7 5 7 3 3 2 2	17 8 1 2 5 0 1 0 0	2.69 1.26 0.16 0.32 0.79 0 0.16 0 0	17 4 4 2 1 1 2 2 1	1.53 0.36 0.36 0.18 0.09 0.09 0.18 0.18 0.09	0.06
**Second primary malignancy** **Non-melanoma skin cancer** **Squamous cell carcinoma** ** Malignant melanoma** **Prostate** **Breast** **Lung** **Gastrointestinal** **Lymphoma** **Other**	52 12 4 2 8 7 4 5 1 9	19 6 0 0 2 4 2 1 1 3	2.88 0.92 0 0 0.30 0.61 0.30 0.15 0.15 0.45	30 5 4 1 5 3 2 4 0 6	2.74 0.45 0.37 0.09 0.45 0.28 0.18 0.37 0 0.55	0.34
**Death**	35	15	1.62	20	1.36	0.59
**Blast Phase**	9	5	0.71	4	0.34	0.14
**Post-PV myelofibrosis**	14	5	0.72	9	0.78	0.45

^1^ Events reported for HU-CR (n. 166) and HU-POOR patients (n. 283) occurred during HU therapy and time was considered from index date to HU discontinuation or last contact. Incidence rates were compared between HU-CR/HU-POOR, using the exact mid-P estimation method.

## Data Availability

The authors confirm that the data supporting the findings of this study are available within the article.

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
