# Peer review of "Predictors of Response to Hydroxyurea and Switch to Ruxolitinib in HU-Resistant Polycythaemia VERA Patients: A Real-World PV-NET Study"

_cancers, 2023, doi:10.3390/cancers15143706_

Round 1

Reviewer 1 Report

This is a really very interesting retrospective assessment of PV patients treated primarily with HU. In the light of new first line therapeutic strategies with ropeginterferon proceeding more and more into first line, we also have a proportion of patients who are not candidates for IFN, therefore the findings of this retrospective analysis are very important and interesting.

The Conclusions are identical with the last passage of the discussion, so I think this part should be revised. 

Author Response

Thank you for the comment. Indeed, the conclusions were also erroneously reported in the discussion. We have deleted the repeated parts from the discussion.

Reviewer 2 Report

This is a very interesting  study evaluating the role of HU and RUX in the selected patients. It is a report form a long period and a large chort of patients. It is important as it reflects real life data, some patients were on low dose HU (only partially effeftive) and others switshed to RUX. This comparison is very valuable.

I have only minor technical problem.

The authors stated that all patients agreed to the study. As it is a retrospective study, some patients died and could not consent. Please clarifiy this section, how this was done.  THis was a retrospective non-inerventional study, so this can be sorted out.

All the other content is acceptable and woth publishing.

None.

Author Response

Thank you for the comment.

For patients currently followed at the experimental center, Informed Consent was obtained as part of one of the visits in the normal care pathway.

For deceased patients, Italian regulations authorize the processing of personal data carried out for scientific research purposes (Gazzetta Ufficiale no. 72 dated 03/26/2012). Therefore, the processing of personal data is considered authorized upon approval of the study by the Ethics Committee.

This important point has been detailed in the appropriate section (Ethical aspects, please find the changes in bold red).
